# Entrained neuronal activity to periodic visual stimuli in the primate striatum compared with the cerebellum

Masashi Kameda[1], Shogo Ohmae[1,2], Masaki Tanaka[1]*

[1]Department of Physiology, Hokkaido University School of Medicine, Sapporo, Japan; [2]Department of Neuroscience, Baylor College of Medicine, Houston, United States

**Abstract** Rhythmic events recruit neuronal activity in the basal ganglia and cerebellum, but their roles remain elusive. In monkeys attempting to detect a single omission of isochronous visual stimulus, we found that neurons in the caudate nucleus showed increased activity for each stimulus in sequence, while those in the cerebellar dentate nucleus showed decreased activity. Firing modulation in the majority of caudate neurons and all cerebellar neurons was proportional to the stimulus interval, but a quarter of caudate neurons displayed a clear duration tuning. Furthermore, the time course of population activity in the cerebellum well predicted stimulus timing, whereas that in the caudate reflected stochastic variation of response latency. Electrical stimulation to the respective recording sites confirmed a causal role in the detection of stimulus omission. These results suggest that striatal neurons might represent periodic response preparation while cerebellar nuclear neurons may play a role in temporal prediction of periodic events.

DOI: https://doi.org/10.7554/eLife.48702.001

## Introduction

When listening to music, we often feel rhythms and sometimes dance to them. It has been suggested that temporal information processing for periodic events differs from that for a single interval, such as pressing a stopwatch at a specific time (*Grahn and Brett, 2007*; *Grube et al., 2010*; *Patel and Iversen, 2014*; *Ross et al., 2018*; *Teki et al., 2011a*; *Teki et al., 2011b*). Previous studies of functional imaging (*Aso et al., 2010*; *Grahn and Brett, 2007*; *Hove et al., 2013*; *Konoike et al., 2012*; *Teki and Griffiths, 2016*) and clinical cases (*Allman and Meck, 2012*; *Cope et al., 2014*; *Schlerf et al., 2007*) have elucidated brain regions relevant to rhythm processing in the frontoparietal cortices as well as in the basal ganglia and the cerebellum. However, as the number of physiological studies in experimental animals are limited (*Cadena-Valencia et al., 2018*; *Lakatos et al., 2008*; *Renoult et al., 2006*), how individual neurons in these regions represent periodic timing remains largely unknown. Furthermore, most of the previous studies used timing tasks that required rhythmic movements, making it difficult to dissociate sensory from motor components in neuronal activity (*Bartolo et al., 2014*; *Merchant et al., 2011*; *Schneider and Ghose, 2012*).

With the aim of clarifying the neuronal mechanism of rhythm processing, we have developed the oddball detection paradigm that requires temporal prediction of periodic events (*Ohmae et al., 2013*). In this task, subjects are asked to detect the unexpected omission or the change in color of isochronous repetitive visual stimuli in the range of several hundreds of milliseconds. Importantly, to detect the stimulus omission, the subjects need to predict the timing of each stimulus in the sequence but are not allowed to make any movements until the omission. When tested in humans, the subjects indeed relied on temporal prediction to detect the stimulus omission for sequences slower than 4 Hz, whereas they detected a slight stumbling of stimulus stream for faster sequences

*For correspondence:
masaki@med.hokudai.ac.jp

**Competing interests:** The authors declare that no competing interests exist.

(*Ohmae and Tanaka, 2016*). We also demonstrated in monkeys that neurons in the cerebellar dentate nucleus during the task exhibited entrainment of activity, and that neuronal modulation for each stimulus was proportional to the inter-stimulus interval (ISI) (*Ohmae et al., 2013*). Because local inactivation and electrical stimulation applied to the recording sites delayed and promoted the detection of the stimulus omission, respectively, the periodic neuronal signals in the cerebellar nucleus likely contributed to the prediction of stimulus timing during the task (*Ohmae and Tanaka, 2016*; *Ohmae et al., 2013*; *Uematsu et al., 2017*).

Nevertheless, the cerebellum is not the only subcortical structure involved in rhythm perception. As above, evidence shows that the basal ganglia are also essential for temporal information processing for both periodic and single event timing (*Buhusi and Meck, 2006*; *Coull et al., 2011*; *Merchant et al., 2013*; *Paton and Buonomano, 2018*). In particular, dopamine signaling in the striatum appears to play a crucial role in regulating the signal flow in the cortico-basal ganglia pathways during timing tasks (*De Corte et al., 2019*; *Emmons et al., 2016*; *Kim and Narayanan, 2019*; *Kunimatsu and Tanaka, 2016*; *Soares et al., 2016*). In this study, we examined neuronal activity in the caudate nucleus during the oddball detection task in monkeys, and then compared the results with those obtained previously from the cerebellar dentate nucleus (*Ohmae et al., 2013*). We show that neurons in the caudate nucleus also exhibit periodic activity for isochronous repetitive visual stimuli, whereas the time courses of neuronal activity were clearly different from those in the cerebellum. Further quantitative analyses suggest that the caudate nucleus might be responsible for periodic motor preparation, while the cerebellar dentate nucleus may be involved in temporal prediction of periodic sensory event which is necessary to detect the stimulus omission.

## Results

### Periodic neuronal activity in the caudate nucleus

A total of 129 task-related neurons were recorded from the head of the caudate nucleus (1–5 mm anterior to the anterior commissure, *Figure 1C*) in three monkeys. Of these, 109 neurons (52, 33 and 24 neurons from monkeys G, H and L, respectively) showed firing modulation for each repetitive visual stimulus, while the remaining neurons only exhibited a phasic activity associated with saccades ($n$ = 15) or reward delivery ($n$ = 3), or showed sustained activity in the middle of the stimulus sequence ($n$ = 2).

*Figure 2A* illustrates the activity of a representative neuron in trials with a 400 ms ISI. This neuron showed virtually no response or slight suppression for the first few stimuli in the sequence, and exhibited a strong firing modulation as the repetition progressed in trials with the stimulus omission (missing condition, red trace and rasters). The firing rate started to increase just before the stimulus presentation (vertical dashed lines) and peaked approximately 130 ms following the stimulus onset. The firing modulation was much lower when the animal prepared for detecting the changes in stimulus color (deviant condition, blue trace and rasters). Like this example, the majority of caudate neurons responding to the repetitive stimuli (74%, 83/109) exhibited a gradual rise in firing modulation for each stimulus during the trial, while only one neuron responded roughly equally to each stimulus in the sequence. On average, the neuronal activity of these increase-type neurons peaked at 160 ± 75 ms (SD, $n$ = 84) from the stimulus onset in trials with a 400 ms ISI. Most of them (88%, 74/84) also exhibited a transient activity around the time of saccades following the omission of regular stimulus (*Figure 2A*, right panel, red trace). However, when we tested formally in the conventional visual saccade trials, the increase-type neurons ($n$ = 3) did not exhibit a strong saccade-related activity, if any.

As with another example shown in *Figure 2B*, a minority of neurons (23%, 25/109) responded vigorously to the initial few stimuli in the sequence and exhibited a gradual decrease of activity that eventually disappeared at the time of oddball (i.e. stimulus omission or deviation). For all these decrease-type neurons, the response to the repetitive stimulus was greatest for the first stimulus, exhibiting a peak at 566 ± 358 ms (SD, $n$ = 25) from the stimulus onset. All but four of these neurons showed no transient activity around the time of saccades, just like the example shown in *Figure 2B* (right panel). As we were interested in the underlying neuronal mechanism of oddball detection in this study, we only considered the increase-type neurons further (*Figure 2A*).

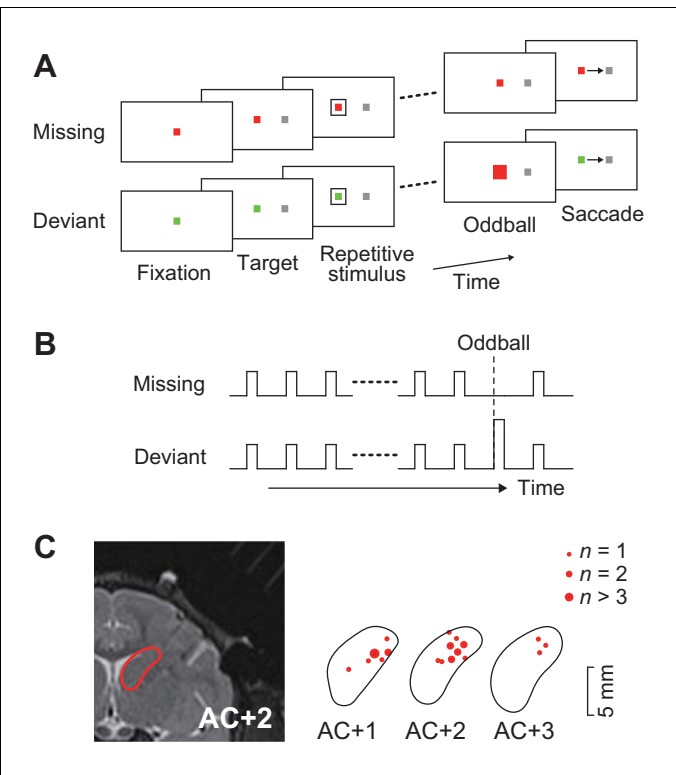

**Figure 1.** Behavioral task and recording sites. (**A**) Oddball detection task. During central fixation, a saccade target was presented either left or right of the fixation point (FP). Then, a brief visual cue (35 ms) was isochronously presented around the FP (unfilled white square). Monkeys were trained to make a saccade in response to the omission or deviation of the repetitive visual stimuli. The deviant stimulus differed in color and filling but had same size as the regular repetitive stimuli. Different FP colors indicated different conditions so that animals could predict either stimulus omission or deviation during fixation. The FP remained visible until end of each trial. (**B**) Time course of events. Repetitive visual stimuli were presented at a fixed inter-stimulus interval (ISI) ranging from 100 to 600 ms. The stimulus sequence lasted for a random 2000–4800 ms before the occurrence of the oddball. Different conditions were pseudorandomly presented in a block of trials. (**C**) Recording sites reconstructed from MR images. Locations of task-related neurons (red dots) are overlaid on the contour of the caudate nucleus on the coronal sections at different levels from the anterior commissure (AC).
DOI: https://doi.org/10.7554/eLife.48702.002

## Basic response properties in different conditions

The magnitude of neuronal activity for the repetitive stimuli depended on the location of the saccade target only in a minority of individual neurons (17%, 14/84, Wilcoxon's rank sum test, p<0.05, 12 and 2 showed a preference for contralateral and ipsilateral targets, respectively, *Figure 3C*, filled circles), while there was a significant contralateral bias in the population (paired *t*-test, $t_{83} = 3.40$, p=0.005). When the magnitude of directional bias was quantified by computing a regression slope in *Figure 3C*, neuronal activity in the missing oddball condition with the contralateral saccade target was approximately 5% greater than that with the ipsilateral saccade target (slope = 1.08). Given the relatively weak directional modulation, we combined the data for both saccade directions for the subsequence analyses.

Each neuron was tested for two oddball conditions presented randomly: the missing condition required temporal prediction while the deviant condition did not. The two conditions were indicated by color of the fixation point at the beginning of each trial (Materials and methods). As seen in *Figure 2*, neurons in the caudate nucleus exhibited greater firing modulation during the missing than the deviant oddball conditions. *Figure 3A* compares the time courses of population activity between the oddball conditions aligned with either the occurrence of stimulus omission (Miss) or color deviation (Dev). In the neuronal population, the activity in the missing condition was again strongly

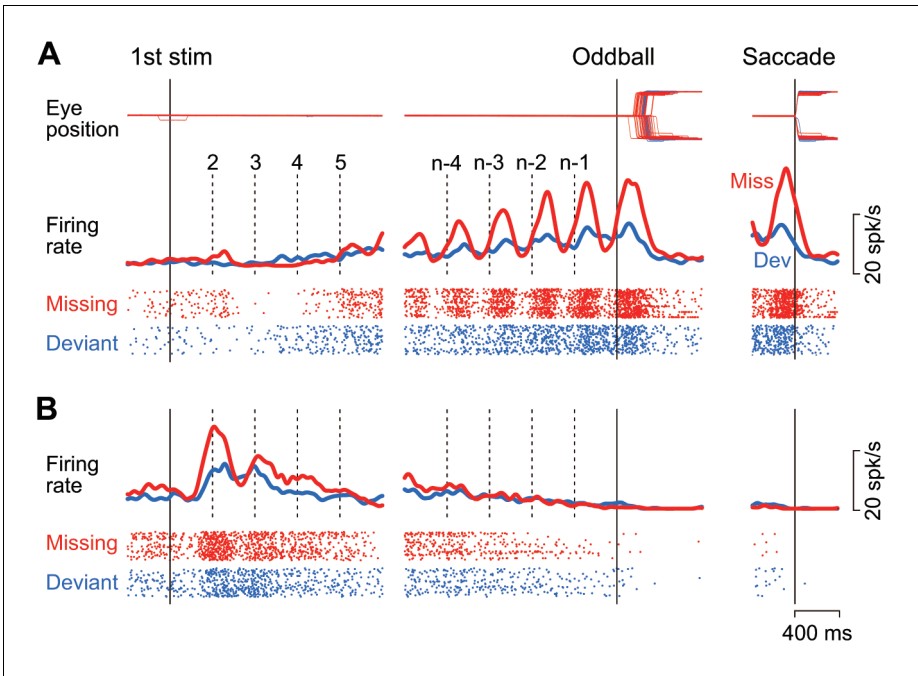

**Figure 2.** Example neurons. (**A**) A neuron with a gradual increase of firing modulation for the repetitive stimuli. Data are aligned with either the first stimulus in the sequence (left) or the occurrence of the oddball (right). Vertical dashed lines indicate stimulus timing. Red and blue traces and rasters indicate the data for the missing and the deviant conditions, respectively. (**B**) A neuron showing a gradual decrease of firing rate.

DOI: https://doi.org/10.7554/eLife.48702.003

The following source data is available for figure 2:

**Source data 1.** Numerical data for *Figure 2*.
DOI: https://doi.org/10.7554/eLife.48702.004

enhanced compared with the deviant condition. When the magnitude of neuronal activity was assessed by measuring the peak (100 ms bin) during 400 ms before the oddball (*Figure 3A*, black bar), 50 out of 84 neurons showed differential activity between the conditions (Wilcoxon's rank sum test, p<0.05, *Figure 3B*, filled circles). In the population as a whole, the activity in the missing condition was approximately 1.7-fold greater than that in the deviant condition (paired *t*-test, $t_{83}$ = 11.7, p<$10^{-19}$). Thus, the periodic activity in the caudate nucleus was much greater when the prediction of stimulus timing was needed. *Figure 3D* plots the time courses of firing modulation for each repetitive stimulus, showing that the peak activity (data connected with solid lines) gradually increased during stimulus repetition while the minimal value during each ISI (100 ms bin, open symbols connected with dashed lines) remained almost unchanged throughout the trial in both oddball conditions.

We previously demonstrated that the periodic neuronal modulation in the cerebellar dentate nucleus depended on the stimulus interval, showing a greater firing modulation for longer ISIs (*Ohmae et al., 2013*; *Uematsu et al., 2017*). In this study, we also found that the magnitude of periodic activity in the caudate nucleus altered as the ISI was systematically varied from 100 to 600 ms. Like neurons in the cerebellum, a representative caudate neuron shown in *Figure 4A* exhibited greater firing modulation for longer ISIs. Among 49 increase-type neurons tested formally, the majority (76%, 37/49) were classified into long-tuned neurons. *Figure 4C* summarizes the normalized firing modulation for these neurons. On average, the normalized activities for the 100, 200, 300 and 400 ms ISIs were 9, 18, 43% and 64% of the response to the 600 ms ISI, respectively.

We also found a subset of caudate neurons (24%, 12/49) exhibiting a preference for the medium ISIs like the one shown in *Figure 4B*. This short-tuned neuron showed the greatest firing modulation for the 300 ms ISI during fixation, as well as a transient activity associated with saccades for short intervals. *Figure 4D* summarizes the normalized firing modulation for different ISIs for this and the

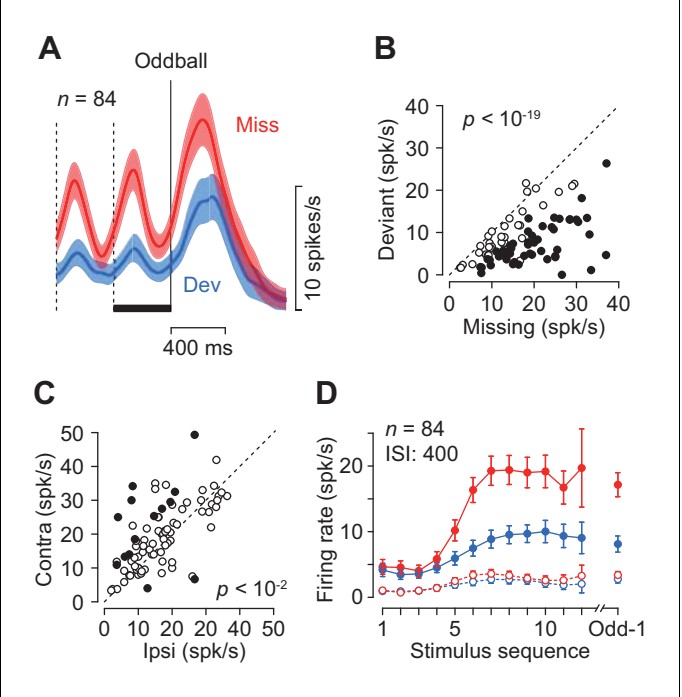

**Figure 3.** Quantitative comparison of neuronal activity between the missing and the deviant conditions. (A) Population activity of the increase-type neurons aligned with the oddball (vertical solid line). Shaded area indicates ± 95% CIs. (B) Effects of oddball conditions in individual neurons. The maximal firing rate was measured during a 100 ms window placed within the last inter-stimulus interval denoted by a black bar in A. Filled symbols indicate the data with a significant difference (Wilcoxon's rank sum test, p<0.01, corrected). (C) Effects of saccade direction. Each data point compares maximal activity before the oddball between trials with opposite saccade directions. (D) The means of maximal (data connected with solid lines) and minimal (dashed lines) activities during each ISI as a function of time. Colors indicate different oddball conditions. Error bar indicates ± 95% CIs. Data are aligned with the beginning of the stimulus sequence (left panel) or the occurrence of the oddball (right).
DOI: https://doi.org/10.7554/eLife.48702.005

The following source data is available for figure 3:

**Source data 1.** Numerical data for *Figure 3*.
DOI: https://doi.org/10.7554/eLife.48702.006

other short-tuned neurons. Almost all neurons (92%, 11/12) exhibited maximal firing modulation for either 300 or 400 ms ISIs, and none showed a preference for the 100 ms ISI. In the population, the magnitude of firing modulation was 15, 46, 88, 94% and 59% of the maximal value for the ISIs of 100, 200, 300, 400 and 600 ms, respectively. The absence of neurons showing a preference for the 100 ms ISI was not attributed to the use of a Gaussian kernel with a relatively large σ value (30 ms, Materials and methods) to obtain spike density. When we used a narrower kernel of 15 ms, the traces of neuronal activity became somewhat noisy and the magnitude of firing modulation became slightly greater, but the time courses of neuronal activity were very similar to those using the 30 ms kernel (*Figure 4—figure supplement 1A and B*). Furthermore, the duration preference remained almost unchanged for Gaussians with a σ of 15 ms (only two long-tuned neurons altered their preference from 600 to 400 or 300 ms, and one short-tuned neuron altered its preference from 400 to 300 ms) (*Figure 4—figure supplement 1C and D*). In this case, the normalized activity for the long-tuned neurons averaged 18, 26, 51, 68% and 99.7% and the short-tuned neurons averaged 23, 58, 88, 91% and 60% of the maximal value for the ISIs of 100, 200, 300, 400 and 600 ms, respectively.

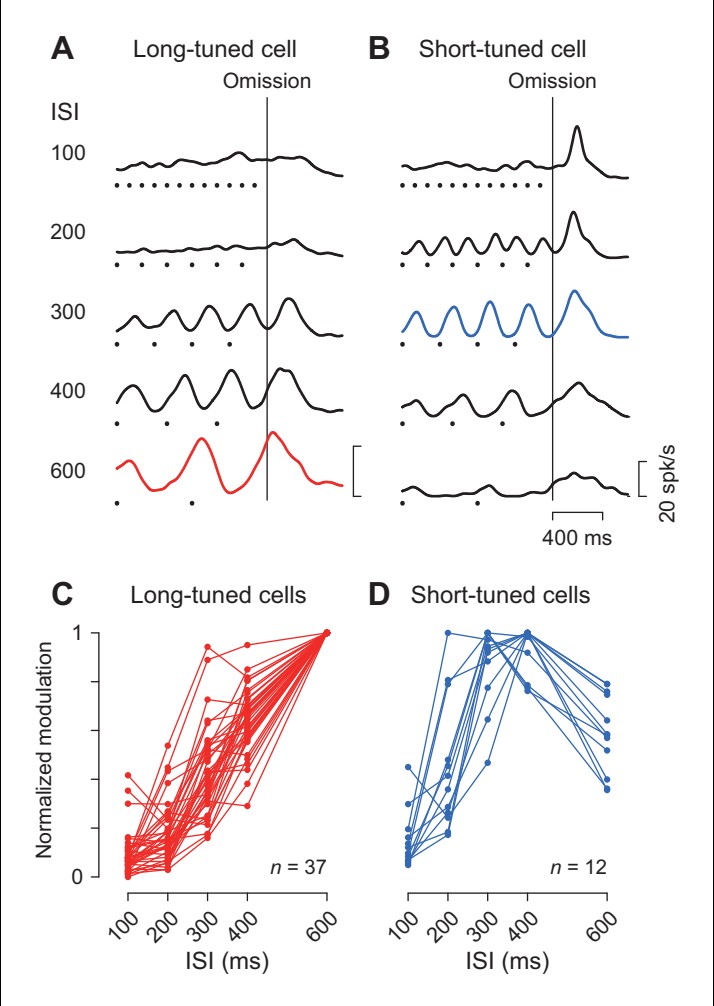

**Figure 4.** Duration preference. (**A, B**) Spike density profiles of two example neurons for different ISIs. Data are aligned with the stimulus omission (vertical black line). Black dots below the traces indicate stimulus timing. Each panel plots a long-tuned neuron (**A**) and a short-tuned neuron (**B**). (**C, D**) Duration tuning for individual neurons. Data points connected with lines indicate the normalized firing modulation for each neuron. Note that the majority of neurons (75%, 36/48) steadily elevated the magnitude of firing modulation as the ISI increased (**C**). Note also that most short-tuned neurons showed a preference for either 300 or 400 ms ISI but never for 100 or 200 ms (**D**).
DOI: https://doi.org/10.7554/eLife.48702.007

The following source data and figure supplement are available for figure 4:

**Source data 1.** Numerical data for *Figure 4*.
DOI: https://doi.org/10.7554/eLife.48702.009
**Figure supplement 1.** Comparison of different Gaussian filters.
DOI: https://doi.org/10.7554/eLife.48702.008

## Comparison of time course of neuronal activity with the deep cerebellar nuclei

Although neurons in both the caudate nucleus and the deep cerebellar nucleus exhibited periodic firing modulation for the isochronous visual stimuli, the time courses of neuronal activity were quite different. *Figure 5A* plots the population activity for the increase-type neurons in the caudate nucleus for different ISIs. In this figure, the data are aligned with the stimulus just before the stimulus omission, and the dashed traces indicate neuronal activity during 100 ms following the stimulus omission. The neuronal firing in the caudate nucleus peaked approximately 160 ms after the stimulus onset and terminated at ~ 300 ms in trials with longer ISIs (300–600 ms). In contrast, neurons in the

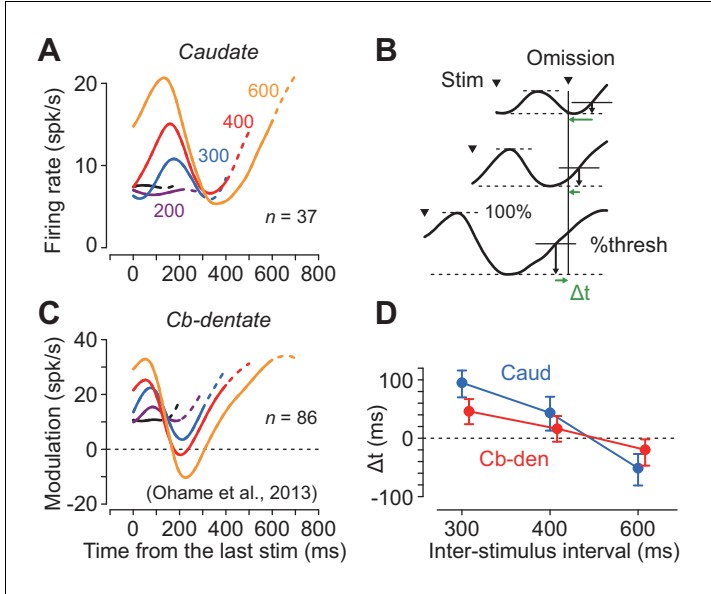

**Figure 5.** Comparison of neuronal activity in the caudate nucleus with that in the cerebellar dentate nucleus. (A, C) Time courses of population activity aligned with the stimulus just before the oddball. Different colors indicate different ISIs. Dashed traces indicate neuronal activity during 100 ms following the oddball. Data for the cerebellar nucleus are taken from the previous study (*Ohmae et al., 2013*). (B) Procedures to estimate the next stimulus timing based on the time courses of the population activity. For the data of longer ISIs (300, 400 and 600 ms), we assumed that the stimulus timing was predicted when neuronal activity surpassed a certain threshold. We searched for the % threshold that minimized the squared sum of the prediction error (Δt) of the next stimulus timing. (D) Prediction error of stimulus timing for neurons in the caudate nucleus and the cerebellum. Error bars indicate bootstrap 95% CIs.

DOI: https://doi.org/10.7554/eLife.48702.010

The following source data and figure supplement are available for figure 5:

**Source data 1.** Numerical data for *Figure 5*.
DOI: https://doi.org/10.7554/eLife.48702.012
**Figure supplement 1.** Effects of prior instruction of oddball condition on neuronal activity in the cerebellar dentate nucleus.
DOI: https://doi.org/10.7554/eLife.48702.011

cerebellar dentate nucleus reported previously (*Ohmae et al., 2013*; *Figure 5C*) exhibited a transient decrement of activity that was lowest approximately 200 ms following the stimulus onset, and gradually resumed the firing rate until the time of the next stimulus. Thus, neurons in the caudate nucleus and the cerebellar dentate nucleus exhibited firing modulation for the repetitive visual stimuli in opposite directions, and the time courses of recovery from the transient activity for each stimulus clearly differed between the recording sites.

During recording from caudate neurons in the present study, color of the fixation point indicated the oddball condition so that the animals expected either stimulus omission or deviation during the stimulus repetition (*Figure 1A*). However, during recording from the deep cerebellar nucleus in the previous study, animals were not informed in advance of oddball conditions so that they were required to predict stimulus timing in all of the trials. To test the effect of prior instruction of oddball condition, we recorded from additional 18 neurons in the cerebellar dentate nucleus in monkey H in separate sessions. We found that the time course of neuronal activity was similar to that in *Figure 5C*, and the times of peak and trough of neuronal activity were not significantly altered by the instruction, although the magnitude of firing modulation differed between these populations of neurons (*Figure 5—figure supplement 1*).

We next examined how precisely neuronal activity in these subcortical structures signaled the timing of each next stimulus. We assumed that the downstream structure might decode the elapsed

time when the firing rate reached a certain threshold that was proportional to the size of firing modulation. First, we measured the maximal firing modulation as the difference between the peak and trough of population activity during the period from the last stimulus to the time of stimulus omission (i.e. the expected time of stimulus appearance) for each ISI. Then, we defined the predicted timing as the time when the firing modulation reached a certain level of the maximal value. We searched for the optimal %threshold (1% steps) across all the ISIs of 300, 400 and 600 ms that minimized the sum of the squared difference between the predicted and actual timing of the next stimulus (temporal prediction error, Δt, *Figure 5B*). *Figure 5D* plots the prediction errors in different interval conditions, showing that the stimulus timing was better predicted by neuronal activity in the cerebellum than the caudate nucleus (mean Δt = 63 versus 28 ms; bootstrap 95% CI [46, 81] and [14 44], respectively). These results suggest that the cerebellum may carry more accurate temporal information than the striatum when predicting timing of periodic events.

The previous study using the oddball detection task demonstrated that neuronal activity in the cerebellar nucleus at the time of stimulus omission correlated with saccade latency (*Ohmae et al., 2013*). The other study in our laboratory using the self-timing task showed that the stochastic variation of saccade latency differently affected neuronal activity in the caudate and the cerebellar dentate nucleus (*Kunimatsu et al., 2018*). Therefore, we next asked whether the timing of neuronal correlates of trial-by-trial latency during the oddball task differed between these subcortical structures. To examine this, data for each neuron were divided into six groups according to saccade latency and were aligned with either stimulus omission or saccades (Materials and methods). Then, we measured the timing of stochastic neuronal variation across different groups by performing ANOVA for every millisecond. Specifically, when we aligned the data with stimulus omission, the traces of population activity for the six latency groups were initially comparable but gradually varied as time passage (*Figure 6A and C*, left panels). We searched for the start of the diverging point back in time, and the time when p-value consistently (>100 ms) surpassed 0.05 was taken as the start of neuronal variation. Similarly, when we aligned the data with saccade initiation, the neuronal activity initially varied between the groups but become indistinguishable around the time of saccades (right panels). We searched for the end of neuronal variation in order of time and detected the time when p-value consistently became > 0.05. In the caudate nucleus, the population activities for different latency groups varied until just before saccade onsets (–161 ms, *Figure 6A*, vertical dashed line), while neuronal activity around the time of saccades (±50 ms) were comparable between the groups (one-way ANOVA, p=0.90). In contrast, the stochastic neuronal variation in the cerebellar nucleus disappeared well before saccade initiation (–428 ms, *Figure 6C*). We evaluated the difference in timing of neuronal variation using the bootstrap procedure and found that when the data were aligned with saccade initiation the trial-by-trial variation disappeared earlier in the cerebellum than the caudate nucleus (*Figure 6D*). When the same sets of data were aligned with the stimulus omission, the stochastic variation tended to emerge earlier in the cerebellum than the caudate nucleus (*Figure 6B*). Thus, neurons in the cerebellar nucleus exhibited trial-by-trial variation around the time of stimulus omission, while those in the caudate nucleus showed stochastic variation just before saccade initiation.

## Causal role of striatal activity in the detection of stimulus omission

To examine the causal role of the caudate neuronal activity in the oddball detection, we delivered electrical microstimulation to the recording sites. *Figure 7A* illustrates eye position traces for contralateral saccades in trials with and without electrical stimulation (100 Hz, 50 μA, color shade). Although electrical stimulation delivered 200 ms before the oddball did not evoke immediate saccades, the detection of stimulus omission was clearly promoted (*Figure 7A*, red traces). The same stimulation pulses did not alter the detection of stimulus deviation (blue traces) in this and most of the other stimulation experiments (see below).

*Figure 7B* plots the medians of contraversive saccade latencies for the stimulus omission in individual experiments. In 14 out of 29 sessions (48%), electrical stimulation significantly altered saccade latency (Wilcoxon's rank sum test, p<0.05 with Bonferroni correction), and the means of median latencies were significantly different in the population (paired *t*-test, $t_{28}$ = 10.2, p<$10^{-11}$). Furthermore, electrical stimulation also reduced the variation of saccade latency for the stimulus omission in the population (SDs averaged 58.0 ± 16.4 ms and 47.1 ± 13.1 ms, paired *t*-test, $t_{57}$ = 3.45, p<0.005), while this effect was not observed in individual experiments (F-test, p>0.05, corrected). In contrast,

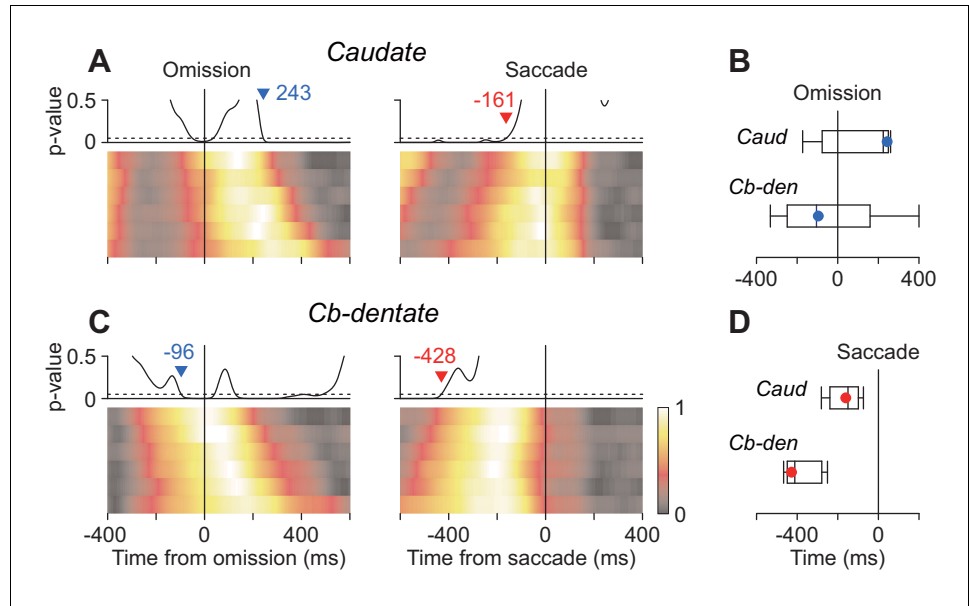

**Figure 6.** Neuronal correlates of stochastic variation of saccade latency. (A, B) For each neuron, data were divided into six groups according to saccade latency (with equal number of trials). Each heatmap plots normalized population activity aligned with either stimulus omission (left panel) or saccade initiation (right). We performed one-way ANOVA for every millisecond to measure the timing of neuronal correlates of trial-by-trial variation in latency (Materials and methods). Traces above the heatmap plot the time course of p-values and the horizontal dashed lines denote the significance level (0.05). Inverted triangles with numbers indicate the timing when neuronal variation consistently (>100 ms) started (blue) or disappeared (red). Note that the stochastic variation started firstly in the cerebellar nucleus (B) and persisted late in the caudate nucleus (A). (C) Summary of the timing of neuronal variation relative to the stimulus omission (top panel) or saccade initiation (bottom). Box-whisker plot illustrates bootstrap quartiles and 95% CIs. Red and blue circles indicate actual measurements.
DOI: https://doi.org/10.7554/eLife.48702.013

The following source data is available for figure 6:

**Source data 1.** Numerical data for *Figure 6*.
DOI: https://doi.org/10.7554/eLife.48702.014

in trials with stimulus deviation, electrical stimulation failed to alter saccade latency in most of individual experiments (2/29, 7%, Wilcoxon's rank sum test, p<0.05, corrected) as well as in the population (paired *t*-test, $t_{28}$ = 1.50, p=0.14, *Figure 7D*). The variation of contraversive saccade latency in the deviant oddball condition was also reduced during electrical stimulation (paired *t*-test, $t_{28}$ = −2.24, p<0.05).

*Figure 7C* summarizes the changes in the medians of reaction time in all experimental conditions. Electrical stimulation facilitated ipsiversive saccades in the missing condition (5 out of 29 sessions, Wilcoxon's rank sum test, p<0.05, corrected; in the population, paired *t*-test, $t_{28}$ = 7.76, p<$10^{-8}$), and also in the deviant condition to a lesser extent (1 out of 29, Wilcoxon's rank sum test, p<0.05, corrected; in the population, paired *t*-test, $t_{28}$ = 6.50, p<$10^{-7}$). Two-way ANOVA revealed significant main effects (oddball condition, $F_{1,84}$ = 118.28, p<0.05; saccade direction, $F_{1,84}$ = 5.59, p<0.05) and interaction ($F_{1,84}$ = 10.96, p<0.05). Post hoc paired *t*-tests detected a significant difference between saccade directions in the missing but not in the deviant conditions ($t_{28}$ = 3.23 and −1.51, p=0.005 and 0.14, respectively). These results indicate that the neuronal signals in the caudate nucleus were involved in making decision in response to a single omission of a regular beat.

## Discussion

We found that neurons in the caudate nucleus exhibited periodic firing modulation when monkeys attempted to detect a single omission of repetitive visual stimulus. Most neurons showed phasic activity that increased gradually as the repetition progressed and peaked around 160 ms following

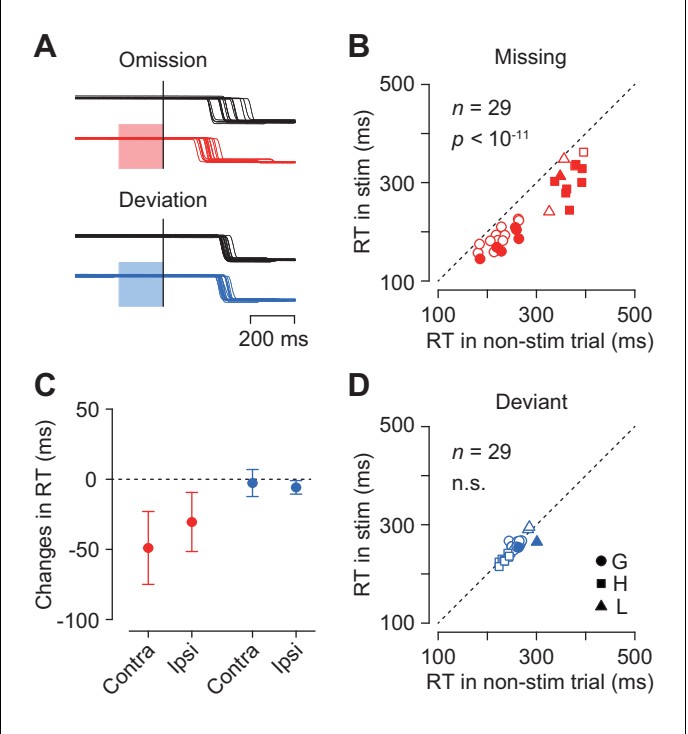

**Figure 7.** Effects of electrical microstimulation. (**A**) Traces of eye position aligned with the stimulus omission or deviation in a single experiment. Black traces indicate non-stimulation controls and colored traces indicate randomly-interleaved stimulation trials. Shaded areas denote timing of electrical stimulation (200 ms, 50 µA, 100 Hz). (**B, D**) Comparison of medians of contraversive saccade latencies between trials with and without electrical stimulation. Different symbols indicate different animals. (**C**) Changes in saccade latency for different oddball conditions and saccade directions.

DOI: https://doi.org/10.7554/eLife.48702.015

The following source data is available for figure 7:

**Source data 1.** Numerical data for *Figure 7*.
DOI: https://doi.org/10.7554/eLife.48702.016

the visual stimulus. The firing modulation for the repetitive stimulus did not develop as great as missing condition when monkeys attempted to detect the change in stimulus color (deviant condition), although the animals were equally required to make a saccade to the visible target. We also found that electrical stimulation delivered to the recording sites promoted the detection of stimulus omission but not the detection of stimulus change. These results suggest that neuronal activity in the caudate nucleus might be relevant to periodic motor preparation with strong enhancement by temporal prediction and may be causally linked with the detection of stimulus omission.

## Entrained neuronal activity in the caudate nucleus

Most (88%) caudate neurons with periodic activity exhibited a strong burst of activity around the time of saccades, suggesting that they could somehow play a role in eye movement generation. In support of this, the magnitude of periodic activity in one fifth of individual neurons as well as in the population of neurons altered depending on saccade direction (*Figure 3C*). It has been widely accepted that an increase of neuronal firing in the caudate nucleus results in a reduction of activity in the substantia nigra, leading to disinhibition of neurons in the superior colliculus which facilitates saccade generation (*Hikosaka et al., 2000*; *Shires et al., 2010*). In fact, electrical stimulation delivered to the caudate nucleus has been shown to evoke contraversive saccades with short latency (typically < 150 ms) (*Kitama et al., 1991*; *Yamamoto et al., 2012*). In this study, however, electrical microstimulation applied to the recording sites failed to elicit immediate saccades but instead reduced the latency of targeting saccades in both directions. Given that the stimulation parameters

in our study was comparable or even stronger than those in the previous studies, the differential stimulation effects might be due to the difference in stimulation sites within the caudate nucleus (i.e. more anterior sites in this study) or the difference in behavioral context (*Watanabe and Munoz, 2013*). Furthermore, the effects of electrical stimulation in many sites were evident only when the animals prepared for saccades in response to the stimulus omission (*Figure 7B–D*). These results suggest that neurons with periodic activity may not directly participate in the triggering of saccades, although they could be related to saccade preparation. Previous studies demonstrated that neuronal firing in the primary motor cortex (*Renoult et al., 2006*) as well as the power of gamma-band oscillation in the supplementary motor area (*Cadena-Valencia et al., 2018*) exhibited periodic modulation in anticipation of the trigger signal for reaching movements. Considering these results together, the temporal prediction of repetitive events might be represented by periodic motor preparation in the cortico-basal ganglia circuitry.

We also found that the periodic activity was much less when monkeys prepared for saccades in response to the changes in stimulus color. Because the repetitive visual stimuli and required movements were identical between the oddball conditions, the different firing modulation was likely attributed to the need for temporal prediction in the missing but not in the deviant conditions. Alternatively, the difference in neuronal activity might come from the fact that there was an explicit cue for the timing of the targeting saccade in the deviant condition, while the animals needed to internally determine movement timing for stimulus absence in the missing condition. In relation to this, previous studies have shown that neuronal firing and stimulation effects in several brain areas are different between self-initiated and triggered movements (*Kunimatsu et al., 2018*; *Kunimatsu and Tanaka, 2012*; *Maimon and Assad, 2006*; *Ohmae et al., 2017*; *Turner and Anderson, 2005*; *van Donkelaar et al., 1999*; *Zimnik et al., 2019*), while significant preparatory activity can be observed when the timing of the trigger signal is certainly predictable (*Romo and Schultz, 1992*; *Tanaka, 2007*).

It is also well documented that neurons in the caudate nucleus modulate firing in association with reward (*Kawagoe et al., 1998*; *Lau and Glimcher, 2007*; *Lauwereyns et al., 2002*; *Schultz, 2016*). However, in this study, monkeys obtained an equal amount of reward for every successful trial. These results indicate that the differential modulation of periodic activity between the conditions unlikely reflected reward information. Nevertheless, the activity of these neurons as well as behavioral parameters could be altered by systematic changes in the reward schedule. For example, the gradual rise of neuronal activity during stimulus repetition (*Figure 3D*), which likely reflected temporal expectation of the stimulus omission in each trial (*Uematsu et al., 2017*), might be scaled by the amount of reward. This possibility is to be tested in future studies.

## Comparison of periodic activity between caudate and cerebellum

Several lines of evidence suggest that different time intervals might be processed by different groups of neurons (*Hayashi et al., 2015*; *Heron et al., 2012*). In support of this hypothesis, neurons in several areas in the cerebral cortex have been shown to exhibit tuned representation of specific intervals during rhythmic movements (*Bartolo et al., 2014*; *Hayashi et al., 2015*; *Heron et al., 2012*; *Merchant et al., 2011*). We also found in this study that a minority of neurons in the caudate nucleus showed a tuned representation for specific time intervals (200–400 ms, *Figure 4B,D*), while the majority of neurons exhibited monotonical increase of firing modulation as a function of ISI. In contrast, tuned representation for short ISIs was not found in the cerebellar dentate nucleus in the previous study (*Ohmae et al., 2013*). These results suggest that the duration tuning might only be common in the cortico-basal ganglia pathways but not in the cortico-cerebellar pathways, although alternative possibilities still remain. For example, we could find tuned representation in the cerebellum if we tested for a wider range of intervals than in the previous study (100–600 ms), or if we exhaustively searched for relevant neurons in the other cerebellar nuclei.

Another line of evidence suggests a scalable neuronal representation of interval timing. When monitoring a single interval, the time courses of neuronal activity in the cerebral cortex and the striatum are temporally scaled according to the duration of the measured interval (*Kunimatsu et al., 2018*; *Mello et al., 2015*; *Wang et al., 2018*; *Xu et al., 2014*). Alteration in the speed of neuronal processing in the population of neurons could be achieved by changing the initial state or the overall level of excitatory input to the recurrent network (*Murray and Escola, 2017*; *Suzuki and Tanaka, 2019*; *Wang et al., 2018*). However, the caudate neurons found in the present study significantly

altered the *magnitude* of firing modulation for different ISIs (*Figure 4*), suggesting that the temporal information processing for periodic events might differ from the scaling representation for single-event timing. Similar to neuronal firing in the caudate and the cerebellar nucleus, temporally specific gain modulation has also been reported in the beta power in the sensorimotor cortices and the neural coherence through the cerebello-cortical pathways during passive listening to isochronous auditory rhythms (*Fujioka et al., 2012*). The temporally specific gain modulation for periodic timing may be an essential framework to understand neural mechanisms of rhythm perception.

Another interesting feature was the different time courses of neuronal activity between the caudate and the cerebellum. Neurons in the caudate nucleus elevated activity for each repetitive stimulus, while those in the cerebellar dentate nucleus reduced activity that gradually resumed and peaked around the time of the next stimulus. When we attempted to decode the timing of each next stimulus from the time course of population activity, the temporal prediction was better for the cerebellum than the caudate nucleus (*Figure 5D*). This was because the duration of transient decrements of activity in the cerebellar nucleus was highly dependent on the ISI, while that of transient increases of activity in the caudate nucleus was not (*Figure 5A,C*).

We also found that neuronal correlates of trial-by-trial variation of saccade latency emerged at different timings in these subcortical structures. When the data were aligned with saccades, the trial-by-trial variation disappeared early in the cerebellum but persisted late in the caudate (*Figure 6D*). When the same sets of data were aligned with stimulus omission, the stochastic variation in the cerebellum started well before but that in the caudate followed the stimulus omission (*Figure 6B*). Given that the reaction time in our behavioral paradigm depends both on the latencies of omission detection and saccade generation, the different timing of stochastic variation may indicate distinct roles of these subcortical structures; the cerebellum might be responsible for the detection of stimulus omission, while the caudate nucleus might contribute to periodic saccade preparation. Because our behavioral paradigm requires both sensory and motor processing, it is difficult to dissociate these components in neuronal activity. However, close comparisons of directionality and time course of neuronal activity suggest that temporal prediction in the caudate nucleus might be represented in motor rather than sensory domain. Similarly, a recent study has shown that the temporal prediction of alternating imaginary targets recruits gamma-bursts in the supplementary motor area, which appears to reflect periodic motor preparation for reaching (*Cadena-Valencia et al., 2018*).

In the present study, we explored signals in the caudate nucleus because our behavioral paradigm required eye movements and because the striatum has been shown to play a role in timing. However, the signals in two other nuclei in the basal ganglia may be also worth exploring during the task. For comparison with the deep cerebellar nucleus, neuronal signals in the output node of the basal ganglia, such as the substantia nigra, need to be clarified in future study. Exploring neuronal signals in the putamen is also important because previous studies suggest its role in timing, especially when somatic movements are involved (*Bartolo et al., 2014*; *Rao et al., 1997*).

In summary, we found that neurons in the caudate nucleus exhibited periodic increases of activity in response to isochronous repetitive visual stimulus. The magnitude of activity depended both on the number of repetitions and the ISI, similar to the activity in the cerebellar dentate nucleus reported previously (*Ohmae et al., 2013*; *Uematsu et al., 2017*). However, unlike neurons in the cerebellum, the duration of firing modulation for each stimulus was almost constant in the caudate nucleus, making it difficult to decode the timing of each next stimulus from the population activity of striatal neurons. In addition, the neuronal correlates of stochastic variations of saccade timing started earlier in the cerebellum than the striatum, suggesting that they might be involved in the temporal prediction of stimulus occurrence and periodic motor preparation, respectively. In future studies, dissociation of sensory from motor components of periodic activity is needed to elucidate different contributions of these subcortical structures.

## Materials and methods

### Animal preparation

Three Japanese monkeys (*Macaca fuscata*; one female and two males, monkeys G, H and L; 6.0–7.6 kg) were used. All experimental protocols were evaluated and approved by the Hokkaido University Animal Care and Use Committee. The basic surgical procedures of animal preparation were identical

to those described previously (*Tanaka, 2007*). Briefly, the animals were implanted with a pair of head holders and an eye coil under general isoflurane anesthesia in separate surgeries. Analgesics were administered during and a few days following each surgery. After full recovery from the surgery, the animals were trained in the oddball detection task (see below) for several months. Then, a recording cylinder was implanted over a trephined hole for electrode penetration targeting the striatum under the same surgical condition. The location of the cylinder was determined based on stereotaxic coordinates and the magnetic resonance (MR) images taken before the surgery. For monkeys G and L, the cylinder was tilted in the coronal plane to allow for electrode penetration roughly perpendicular to the internal capsule (40° off-vertical, centered at AC+3). For monkey H, the cylinder was installed vertically over both hemispheres (AC+3). Antibiotics were administered during the course of the experiments as necessary.

## Visual stimuli and behavioral tasks

Animals were placed in a darkened booth during the experiments. Visual stimuli were presented on either a 24-inch cathode ray tube monitor (GDM-FW900, Sony; 64 × 44° of visual angle; 60 Hz) or a 27-inch liquid crystal display monitor (XL2720-B, BenQ; 73 × 46°; 144 Hz). Stimulus presentation and data acquisition were controlled by the TEMPO system (Reflective Computing). Five different visual stimuli were presented during the experiments. Red and green filled squares (0.5°) served as the fixation point. A gray-filled square (1.0°) was used as a saccade target. A large (4.0°) white unfilled square surrounding the fixation point was presented repeatedly in each trial, and the color of the square suddenly altered (red filled) in the deviant oddball trials (see below).

We used two versions of the oddball detection task, in which animals were required to detect the unexpected omission (missing condition) or the change in color and filling (deviant condition) of the repetitive visual stimuli. Each trial started with central fixation, and the color of the fixation point indicated the trial type (red and green for the missing and deviant conditions, respectively). As the animals maintained fixation for 2000 ms, a saccade target appeared either 16° left or right of the fixation point. After a random 100–400 ms, a brief (35 ms) visual stimulus was repeatedly presented around the fixation point at a fixed inter-stimulus interval (ISI) of 100, 200, 300, 400 or 600 ms (only 400 ms for the deviant condition). The stimulus sequence lasted for a random 2000–4800 ms (3000–4800 ms for trials with a 600 ms ISI) before the occurrence of stimulus omission or color deviation (we refer to these events as 'oddball'). The central fixation point remained visible until the end of each trial. Animals were trained to detect the oddball and make a saccade to the target within 600 ms. It should be noted that the animals needed to predict the timing of every upcoming stimulus to detect the stimulus omission but not for the detection of stimulus deviation, at least for ISIs longer than several hundreds of milliseconds (*Ohmae and Tanaka, 2016*). Because the trial type was explicitly denoted by the color of the fixation point, the animals were likely to attend either to the stimulus omission or deviation during the stimulus repetition. Correct response was reinforced by a liquid reward delivered at the end of each trial.

## Physiological procedures

A single tungsten microelectrode (Alfa Omega Engineering) was lowered into the anterior part of the caudate nucleus through the 23-gauge stainless steel tube that was guided by the grid system (Crist Instrument Co.) attached to the cylinder. The electrode was advanced using a micromanipulator (MO-972A; Narishige) in 10 μm steps. Neuronal signals were amplified (Model 1800; A-M Systems), filtered (Model 3625; NF Co.), and monitored online using oscilloscopes and an audio device. We could easily locate neurons in the caudate nucleus based on the depth of electrodes, relatively wide action potentials, and their low baseline firing rate. Once we encountered task-related neurons, we attempted to isolate the waveform of single neurons online, using software with real-time template-matching algorithms (ASD; Alpha Omega Engineering). In this study, we did not include data from putative tonically active cholinergic interneurons which are known to exhibit characteristic firing pattern and wider action potentials (*Aosaki et al., 1995*). Neurons included for the analysis had relatively low baseline firing rate and were considered as medium-spiny projection neuron and some GABAergic interneurons. The spike timing data were saved in files along with the data of eye position and visual stimuli during experiments. We collected the data for the trials when monkeys remained fixation > 1500 ms from the first stimulus. Eye position data were directly obtained from

the eye coil device (MEL-25; Enzaishi Kogyo) and were sampled at 1 kHz with 16-bit resolution. The data were later analyzed off-line using MATLAB (Mathworks). During recording sessions (*n* = 84), animals sometimes made an early erroneous targeting saccade (11.3 ± 9.3% and 2.7 ± 3.6% for the missing and deviant conditions, respectively) and rarely failed to detect the oddball (2.2 ± 3.5% and 0.9 ± 2.4%). We considered only correct trials for the quantitative analysis.

After the termination of the recording experiments, we examined the effects of electrical microstimulation applied to the recording sites in two monkeys (G and L). Stimulation pulses (50 μA, 200 μs biphasic) were generated by using a stimulator connected with an isolator unit (SEN-3401 and SS-203J; Nihon Kohden). Electrical stimulation was delivered as a train of 100 Hz pulses during 200 ms just before the oddball. Current intensity was monitored by measuring the voltage across the 1 kΩ resistor placed in series with the electrode.

### Data analysis

When we examined the effects of saccade direction and task condition (*Figure 3*), the magnitude of neuronal activity was assessed by direct spike count during the ISI just before the oddball. The maximal neuronal activity was defined as the mean firing rate for 100 ms during the period. For the analyses in *Figure 4*, quantification of neuronal activity was made based on the modulation of spike density, which was obtained by convolving the millisecond-to-millisecond probability of spike occurrence with Gaussians ($\sigma$ = 30 ms). We also used a smaller $\sigma$ (15 ms) to verify the absence of high-frequency neuronal oscillation for short ISIs (see the text associated with *Figure 4*). The magnitude of firing modulation for different ISIs was defined as the difference between the maximum and minimum values of spike density during a specific period in the trial.

We compared the time courses of neuronal activity between neurons in the caudate nucleus and the cerebellar dentate nucleus (*Figures 5* and *6*). When we estimated how accurately neuronal activities could predict the next stimulus timing (*Figure 5A and D*), we searched for the %threshold that minimized the prediction error of stimulus timing in trials with longer ISIs (300–600 ms). When we measured the timing of neuronal correlates of trial-by-trial latency variation (*Figure 6*), we divided the data for each neuron into six groups according to saccade latency and performed one-way ANOVA for every millisecond to detect the diverging point of spike density profiles. Saccade latency was defined as the time from oddball (either omission or color change) to saccade initiation, which was detected when angular eye velocity exceeded 60°/s immediately after the oddball occurrence. The time when the traces of population activities for the six latency groups consistently differed for > 100 ms was taken as the start and end of the stochastic variation. For both analyses on the population activity, statistical significance was evaluated using the bootstrap procedures (1000 resampling of individual neurons).

For comparison between conditions, Wilcoxon's rank-sum test and paired *t*-test were used, respectively, for individual neuronal activity and the population activity. Effects of electrical microstimulation were also evaluated using two-way ANOVA (*Figure 7C*). Further details of statistical tests are reported in the relevant text.

### Acknowledgements

The authors thank T Mori, A Hironaka, and H Miyaguchi for their assistance with animal care and surgery; M Suzuki and M Saito for administrative help; M Takei and M Kusuzaki at the Research Institute for Electronic Science for manufacturing some equipment; and all lab members for comments and discussions. This work was supported partly by grants from the Ministry of Education, Culture, Sports, Science and Technology of Japan (17H03539, 18H04928, 18H05523, 18J20197) and the Takeda Science Foundation. MK received a fellowship from the Japan Society for the Promotion of Science. Animals were provided by the National Bio-Resource Project.

## Additional information

### Funding

| Funder | Grant reference number | Author |
|---|---|---|
| Ministry of Education, Culture, Sports, Science, and Technology | 17H03539 | Masaki Tanaka |
| Takeda Science Foundation | | Masaki Tanaka |
| Japan Society for the Promotion of Science | | Masashi Kameda |
| Ministry of Education, Culture, Sports, Science, and Technology | 18H04928 | Masaki Tanaka |
| Ministry of Education, Culture, Sports, Science, and Technology | 18H05523 | Masaki Tanaka |
| Ministry of Education, Culture, Sports, Science, and Technology | 18J20197 | Masaki Tanaka |

The funders had no role in study design, data collection and interpretation, or the decision to submit the work for publication.

### Author contributions

Masashi Kameda, Data curation, Formal analysis, Validation, Visualization, Writing—original draft, Writing—review and editing; Shogo Ohmae, Data curation, Writing—review and editing; Masaki Tanaka, Conceptualization, Supervision, Funding acquisition, Validation, Writing—original draft, Writing—review and editing

### Author ORCIDs

Masashi Kameda (ID) https://orcid.org/0000-0003-0201-8493
Shogo Ohmae (ID) https://orcid.org/0000-0003-1726-4961
Masaki Tanaka (ID) https://orcid.org/0000-0002-6177-1314

### Ethics

Animal experimentation: All experimental protocols were evaluated and approved by the Hokkaido University Animal Care and Use Committee (#18-0003).

### Decision letter and Author response

Decision letter https://doi.org/10.7554/eLife.48702.019
Author response https://doi.org/10.7554/eLife.48702.020

## Additional files

### Supplementary files

• Transparent reporting form
DOI: https://doi.org/10.7554/eLife.48702.017

### Data availability

Source data files (in Matlab's MAT format) containing numerical data sufficient to reconstruct each main figure have been provided.

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
