## [Decision Letter]

Thank you for submitting your article "Entrained neuronal activity to periodic visual stimuli in the primate striatum compared with the cerebellum" for consideration by *eLife*. Your article has been reviewed by two peer reviewers, including Jennifer L Raymond as the Reviewing Editor and Reviewer #1, and the evaluation has been overseen by Joshua Gold as the Senior Editor.

The reviewers have discussed the reviews with one another and the Reviewing Editor has drafted this decision to help you prepare a revised submission.

Summary:

This is an interesting study, analyzing the neural activity in the basal ganglia and cerebellum of monkeys during a rhythmic timing prediction task. The focus of the present study is recording from the caudate nucleus, with comparison to previously published data from this lab from the dentate nucleus of the cerebellum. In both regions, there is neural activity that predicts the occurrence of the next stimulus, but the neural activity in the cerebellum can better predict the timing of an omitted stimulus than activity in the caudate, whereas activity in the caudate better predicts the behavioral response (saccade) latency. In addition, the authors show that stimulation of the caudate reduces the response latency on the task that requires the monkey to detect stimulus omission, but not a control task that also has rhythmic visual stimuli, but requires the monkey to detect a color change. The results are clear, data rigorously analyzed, and thoughtfully discussed. The many strengths of this study include the comparison of basal ganglia with cerebellum, and the comparison of the stimulus omission and color change tasks, and the convergent evidence from recording and stimulation. It is pretty amazing that differences in neural activity and effects of stimulating the caudate on the two tasks can be detected, especially since the monkeys are trained on both, and hence are likely to be doing some temporal prediction even in the color change task.

Essential revisions:

1) Striatum or caudate.

Are all neurons recorded in caudate (Introduction, last paragraph)? If so, why are they called striatum everywhere else in the manuscript? Are there some neurons outside of the caudate but within the striatum? Unless there are specific reasons, we would suggest using "caudate" consistently.

2) Identify the caudate neurons.

- Are these neurons recorded from the caudate head, body, or tail?

- Have these neurons, or the neurons of the recorded area, been known by some other function in the previous studies? Or, is this brand-new population of neurons that have never tested before?

- Previous studies induced saccades by an electrical stimulation but not in this study (subsection “Entrained neuronal activity in the striatum”, first paragraph). Why is that? Different area? Different stimulation condition?

- How were the neurons determined as caudate dopamine neurons? Criteria needs to be documented.

3) Inconsistency of conclusion.

The conclusion has been mentioned in three locations: (i) at the end of the Introduction; (ii) early part of the Discussion (first paragraph); and (iii) late part of the Discussion (subsection “Comparison of periodic activity between the striatum and cerebellum”, fourth paragraph). It stated that the striatum is for motor preparation at (i) and (iii) but then for temporal prediction at (ii). It stated that the cerebellum is involved in the prediction (Introduction). Which one is true? Maybe the sentences in the first paragraph of the Discussion need to be revised?

4) Comparing with the dentate nucleus data.

- If we understand correctly, the oddball paradigms in this study and the previous dentate study (Ohmae et al., 2013) are slightly different: the target colors during fixation for missing and deviant are different in this study but same in the dentate study. So, the monkey was ready for missing in this study but not in the dentate study. Could this difference cause the difference in the neural activity?

- Dentate nucleus contains output neurons of the cerebellum, whereas the striatum is considered to be an entrance of the basal ganglia. The rationale to compare the dentate nucleus and striatum is not clear. We understand that dentate activity can be considered as a representative of the cerebellum because it is an output of the cerebellum, but striatum cannot. For example, comparing the dentate and substantia nigra is straightforward because both are outputs of the cerebellum and basal ganglia, respectively (subsection “Entrained neuronal activity in the striatum”, first paragraph).

5) Specific concerns:

Subsection “Visual stimuli and behavioral tasks”, last paragraph. In the deviant condition, does the fixation target change only its color? Does the size also change? The size looks changed in Figure 1A.

Subsection “Visual stimuli and behavioral tasks”, last paragraph. What is the visual stimuli around the fixation point? Black square with 6 radial lines as in Figure 1A?

Subsection “Data analysis”, first paragraph. Gaussian 30ms seems long. Why did you choose it? Any reference to support it? If you need to use 15ms for short ISI, why don't you use 15ms for all data? For dentate data, did you use 30ms?

Subsection “Data analysis”, second paragraph. How did you detect the saccade latency?

Figure 1A. At the oddball, should the red fixation point in the missing condition disappear? Also, at the saccade, is the fixation point visible?

Discussion, first paragraph. The activity was not "reduced" in the deviant condition, it did not develop as great as missing condition. This sentence needs to be revised.

Subsection “Entrained neuronal activity in the striatum”, end of first paragraph. This description about previous studies has not been tied to the previous sentences. Why is this here? What are you trying to discuss here? Are you trying to say these are the source signals of the striatum activity?

Subsection “Comparison of periodic activity between the striatum and cerebellum”, first sentence. Again, it's unclear why this sentence is here. How do these studies relate to your result?

Subsection “Comparison of periodic activity between the striatum and cerebellum”, end of fourth paragraph. Another one here again. What are you trying to discuss here?

Subsection “Comparison of time course of neuronal activity with the deep cerebellar nuclei”, second paragraph. Please clarify how modulation of firing was calculated (as a percentage of baseline, or raw delta).

Figure 6. Additional explanation of the analysis and results illustrated in this figure should be added as it is currently a little difficult to interpret.

---

## [Author Response]

Essential revisions:1) Striatum or caudate.Are all neurons recorded in caudate (Introduction, last paragraph)? If so, why are they called striatum everywhere else in the manuscript? Are there some neurons outside of the caudate but within the striatum? Unless there are specific reasons, we would suggest using "caudate" consistently.

All striatal neurons reported here were recorded from the caudate nucleus. We now use "caudate" rather than "striatum" throughout the manuscript, except when we compare the data with previous studies in rodents. We would like to retain the term in the title for contrasting with the cerebellum and for a wide range of readers. We have added a paragraph to Discussion describing why we collected data from the caudate nucleus and that exploration of signals in the putamen in the future study is needed (subsection “Comparison of periodic activity between caudate and cerebellum”, fifth paragraph).

2) Identify the caudate neurons.- Are these neurons recorded from the caudate head, body, or tail?

As stated in the beginning of the Results (subsection “Periodic neuronal activity in the caudate nucleus”, first paragraph), neurons were recorded from the head of the caudate nucleus (1-5 mm anterior to the anterior commissure). We have also added this information to the Materials and methods "anterior part of the caudate nucleus".

- Have these neurons, or the neurons of the recorded area, been known by some other function in the previous studies? Or, is this brand-new population of neurons that have never tested before?

We searched for caudate neurons modulating activity during the oddball detection task, while neurons in this area have been examined using a variety of oculomotor/cognitive tasks in the previous studies. For a few task-related neurons, we also examined activity during a conventional, visual saccade task, but failed to find significant firing modulation. This is mentioned in the revised text (subsection “Periodic neuronal activity in the caudate nucleus”, second paragraph). So, neurons with periodic firing modulation during the oddball task might be a brand-new population, while some of them might also respond to other oculomotor or cognitive tasks tested previously in the head of the caudate nucleus. We have added some note to the text regarding this issue (subsection “Entrained neuronal activity in the caudate nucleus”, first paragraph).

- Previous studies induced saccades by an electrical stimulation but not in this study (subsection “Entrained neuronal activity in the striatum”, first paragraph). Why is that? Different area? Different stimulation condition?

In the previous studies, electrical stimulation applied to the caudate successfully evoked saccades with short latency. Given that the stimulation parameters in our study was comparable to those in the previous studies, the stimulation areas might differ between the studies. However, because the effects of electrical stimulation in the caudate have been shown to be highly context-dependent (e.g., Watanabe and Munoz, 2010), the absence of stimulation-evoked saccades might reflect the difference in task condition rather than the stimulation sites. We have added a sentence regarding the issue: "The differential stimulation effects might be due to the difference in stimulation sites within the caudate nucleus (i.e., more anterior sites in this study) or the difference in behavioral context".

- How were the neurons determined as caudate dopamine neurons? Criteria needs to be documented.

We did not include putative cholinergic, tonically active neurons with characteristic tonic firing pattern and wide action potentials (Aosaki et al., 1995). Neurons included for the analysis had relatively low baseline firing rate and were considered as medium-spiny projection neurons and some GABAergic interneurons. We now mention this point in the revised Materials and methods (subsection “Physiological procedures”, first paragraph).

3) Inconsistency of conclusion.The conclusion has been mentioned in three locations: (i) at the end of the Introduction; (ii) early part of the Discussion (first paragraph); and (iii) late part of the Discussion (subsection “Comparison of periodic activity between the striatum and cerebellum”, fourth paragraph). It stated that the striatum is for motor preparation at (i) and (iii) but then for temporal prediction at (ii). It stated that the cerebellum is involved in the prediction (Introduction). Which one is true? Maybe the sentences in the first paragraph of the Discussion need to be revised?

In response to this comment, we have revised the text in the first paragraph of Discussion (corresponding to the part (ii) indicated by the reviewer). We assume that the periodic activity in the caudate nucleus is relevant to periodic motor preparation and is greatly enhanced by temporal prediction.

4) Comparing with the dentate nucleus data.- If we understand correctly, the oddball paradigms in this study and the previous dentate study (Ohmae et al., 2013) are slightly different: the target colors during fixation for missing and deviant are different in this study but same in the dentate study. So, the monkey was ready for missing in this study but not in the dentate study. Could this difference cause the difference in the neural activity?

The reviewers are right. For the previous recording from the cerebellum, animals were not informed of oddball condition (i.e., missing or deviant) during the stimulus repetition so that they needed to predict stimulus timing in all trials. This is contrast to the present study in which monkeys needed to predict stimulus timing only in the missing trials with a red fixation point (Figure 1A). To examine the effects of prior instruction on neuronal activity, we recorded from 18 additional neurons in the cerebellar dentate nucleus in the new version of the task (i.e., different color of fixation point for different conditions). As shown in Figure 5—figure supplement 1, the magnitude of firing modulation during expectation of stimulus deviation was less than that during expectation of stimulus omission (red continuous versus thin black traces). Nevertheless, the time course of neuronal activity in the missing oddball condition was similar to that presented in Figure 5B (red continuous versus dashed traces), and the times of peak and trough of neuronal activity were almost identical, although the sizes of firing modulation were somewhat different between the two populations of neurons. We have added a paragraph reporting this fact to the Results (subsection “Comparison of time course of neuronal activity with the deep cerebellar nuclei”, second paragraph).

- Dentate nucleus contains output neurons of the cerebellum, whereas the striatum is considered to be an entrance of the basal ganglia. The rationale to compare the dentate nucleus and striatum is not clear. We understand that dentate activity can be considered as a representative of the cerebellum because it is an output of the cerebellum, but striatum cannot. For example, comparing the dentate and substantia nigra is straightforward because both are outputs of the cerebellum and basal ganglia, respectively (subsection “Entrained neuronal activity in the striatum”, first paragraph).

We agree with the reviewers in that exploring neuronal signals in the output nodes of the basal ganglia (the substance nigra and the globus pallidus) is certainly important. However, because many previous studies suggest a role of the striatum in timing, we started from the striatum to compare the signals in the basal ganglia with those in the cerebellum. We have added a paragraph to the Discussion to mention this (subsection “Comparison of periodic activity between caudate and cerebellum”, fifth paragraph).

5) Specific concerns:Subsection “Visual stimuli and behavioral tasks”, last paragraph. In the deviant condition, does the fixation target change only its color? Does the size also change? The size looks changed in Figure 1A.

Only color and filling of the repetitive stimulus changed. We have added a sentence to Figure 1 legend "The deviant stimulus differed in color and filling but had the same size as the regular repetitive stimuli.").

Subsection “Visual stimuli and behavioral tasks”, last paragraph. What is the visual stimuli around the fixation point? Black square with 6 radial lines as in Figure 1A?

The lines represented flashing of repetitive stimuli and were invisible. In response to this comment, we have removed the radial lines.

Subsection “Data analysis”, first paragraph. Gaussian 30ms seems long. Why did you choose it? Any reference to support it? If you need to use 15ms for short ISI, why don't you use 15ms for all data. For dentate data, did you use 30ms?

Because the previous studies in the cerebellar dentate nucleus used a 30-ms Gaussian kernel (Ohmae et al., 2013; Uematsu et al., 2017), we chose the same kernel for comparison. Although the use of a wider kernel smoothened the traces of spike density and reduced the magnitude of firing modulation, there was no qualitative difference from the data with a narrower kernel. To directly compare these data, we have added Figure 4—figure supplement 1 and modified relevant text (subsection “Basic response properties in different conditions”, last paragraph).

Subsection “Data analysis”, second paragraph. How did you detect the saccade latency?

Saccade onset was detected when angular eye velocity exceeded 60 deg/s immediately after the oddball occurrence. Saccade latency was defined as the time from oddball to saccade onset. We now report this in the revised Materials and methods (subsection “Data analysis”, second paragraph).

Figure 1A. At the oddball, should the red fixation point in the missing condition disappear? Also, at the saccade, is the fixation point visible?

Fixation point was visible until the end of trial. We have added this information to the text (subsection “Visual stimuli and behavioral tasks”, last paragraph) and Figure 1 legend.

Discussion, first paragraph. The activity was not "reduced" in the deviant condition, it did not develop as great as missing condition. This sentence needs to be revised.

We have modified the sentences as suggested (Discussion, first paragraph).

Subsection “Entrained neuronal activity in the striatum”, end of first paragraph. This description about previous studies has not been tied to the previous sentences. Why is this here? What are you trying to discuss here? Are you trying to say these are the source signals of the striatum activity?

The source of temporal prediction signals remains unclear. We have added a sentence to the end of this paragraph to clarify the point: "Considering these results together, the temporal prediction of periodic events might be represented by periodic motor preparation in the cortico-basal ganglia circuitry."

Subsection “Comparison of periodic activity between the striatum and cerebellum”, first sentence. Again, it's unclear why this sentence is here. How do these studies relate to your result?

We have added some sentences to make the point clearer: "Several lines of evidence suggest that different time intervals might be processed by different groups of neurons (Hayashi et al., 2015; Heron et al., 2012). […] We also found in this study that a minority of neurons in the caudate nucleus showed a tuned representation for specific intervals (200–400 ms, Figure 4B, D), while the majority of neurons exhibited monotonical increase of firing modulation as a function of ISI."

Subsection “Comparison of periodic activity between the striatum and cerebellum”, end of fourth paragraph. Another one here again. What are you trying to discuss here?

We have added some lines to clarify the point: "Because our behavioral paradigm requires both sensory and motor processing, it is difficult to dissociate these components in neuronal activity. However, close comparisons of directionality and time course of neuronal activity suggest that temporal prediction in the caudate nucleus might be represented in motor rather than sensory domain.".

Subsection “Comparison of time course of neuronal activity with the deep cerebellar nuclei”, second paragraph. Please clarify how modulation of firing was calculated (as a percentage of baseline, or raw delta).

The maximal firing modulation was defined as "*raw data*" which was the difference between the peak and trough of population activity following the last stimulus in the sequence. The predicted stimulus timing was when firing modulation reached a certain threshold that was a percentage of the maximal firing modulation. We have added a few sentences to clarify the details (subsection “Comparison of time course of neuronal activity with the deep cerebellar nuclei”, third paragraph).

Figure 6. Additional explanation of the analysis and results illustrated in this figure should be added as it is currently a little difficult to interpret.

We have inserted some sentences to the text (subsection “Comparison of time course of neuronal activity with the deep cerebellar nuclei”, last paragraph).